# Histologic and Histomorphometric Evaluation of a New Bioactive Liquid BBL on Implant Surface: A Preclinical Study in Foxhound Dogs

**DOI:** 10.3390/ma14206217

**Published:** 2021-10-19

**Authors:** Eduard Ferrés-Amat, Ashraf Al Madhoun, Elvira Ferrés-Amat, Saddam Al Demour, Mera A. Ababneh, Eduard Ferrés-Padró, Carles Marti, Neus Carrio, Miguel Barajas, Maher Atari

**Affiliations:** 1Oral and Maxillofacial Surgery Service, Hospital HM Nens, HM Hospitales, 08009 Barcelona, Spain; eduard.fa@institutferresamat.com (E.F.-A.); elvira.fa@institutferresamat.com (E.F.-A.); eduard.fp@institutferresamat.com (E.F.-P.); 2Department of Animal and Imaging Core Facilities, Dasman Diabetes Institute, Dasman 15462, Kuwait; ashraf.madhoun@dasmaninstitute.org; 3Oral and Maxillofacial Surgery Department, Universitat Internacional de Catalunya, St Josep Trueta s/n, Sant Cugat del Vallès, 08195 Barcelona, Spain; neuscarriober@gmail.com; 4Department of Special Surgery/Division of Urology, School of Medicine, The University of Jordan, Amman 11942, Jordan; saldemour@ju.edu.jo; 5Department of Clinical Pharmacy, Faculty of Pharmacy, Jordan University of Science and Technology, Irbid 22110, Jordan; mababneh@just.edu.jo; 6Biointelligent Technology Systems SL, Diputaccion 316, 3D, 08009 Barcelona, Spain; martipages.c@gmail.com (C.M.); miguel.barajas@unavarra.es (M.B.); 7Oral and Maxillofacial Surgery Department, Hospital Clinic de Barcelona, 08036 Barcelona, Spain; 8Biochemistry Area, Department of Health Science, Public University of Navarre, 31008 Pamplona, Spain; 9Ziacom Medical SL, C. Buhos, 2, 28320 Madrid, Spain

**Keywords:** BLT-SLA active, BBL, dental implant, osseo integration dental implantation

## Abstract

Background: Bioactive chemical surface modifications improve the wettability and osseointegration properties of titanium implants in both animals and humans. The objective of this animal study was to investigate and compare the bioreactivity characteristics of titanium implants (BLT) pre-treated with a novel bone bioactive liquid (BBL) and the commercially available BLT-SLA active. Methods: Forty BLT-SLA titanium implants were placed in in four foxhound dogs. Animals were divided into two groups (n = 20): test (BLT-SLA pre-treated with BBL) and control (BLT-SLA active) implants. The implants were inserted in the post extraction sockets. After 8 and 12 weeks, the animals were sacrificed, and mandibles were extracted, containing the implants and the surrounding soft and hard tissues. Bone-to-implant contact (BIC), inter-thread bone area percentage (ITBA), soft tissue, and crestal bone loss were evaluated by histology and histomorphometry. Results: All animals were healthy with no implant loss or inflammation symptoms. All implants were clinically and histologically osseo-integrated. Relative to control groups, test implants demonstrated a significant 1.5- and 1.7-fold increase in BIC and ITBA values, respectively, at both assessment intervals. Crestal bone loss was also significantly reduced in the test group, as compared with controls, at week 8 in both the buccal crests (0.47 ± 0.32 vs 0.98 ± 0.51 mm, *p* < 0.05) and lingual crests (0.39* ± 0.3 vs. 0.89 ± 0.41 mm, *p* < 0.05). At week 12, a pronounced crestal bone loss improvement was observed in the test group (buccal, 0.41 ± 0.29 mm and lingual, 0.54 ± 0.23 mm). Tissue thickness showed comparable values at both the buccal and lingual regions and was significantly improved in the studied groups (0.82–0.92 mm vs. 33–48 mm in the control group). Conclusions: Relative to the commercially available BLT-SLA active implants, BLT-SLA pre-treated with BBL showed improved histological and histomorphometric characteristics indicating a reduced titanium surface roughness and improved wettability, promoting healing and soft and hard tissue regeneration at the implant site.

## 1. Introduction

Dental implant is a common medical practice to rehabilitate partial or total edentulous patients efficiently, independent of the jawbone quality [1] or regeneration capacity [2], and can be applied immediately after tooth extraction sites [3]. Optimal therapeutic achievement of osseointegration is dependent on the bone density, deployed surgical techniques, implant design, and surface treatment. A combination of these factors influence the primary stability and viability of an implant after surgery [4,5]. The interdigitation degree between the cement-line matrix and implant is reliant on the interface between the implants’ probe and surface reactivity, thus the bone bonding phenomenon is dependent on the bone–bioactive implant interface [6].

Several approaches have been applied to augment the bone-to-implant contact (BIC). A recent study revealed that UV irradiation of Korean implants prior to implantation in dogs did not improve the mean insertion torque or the BIC values in all studied groups at 4 and 12 weeks post implantation [7]. Alternatively, resorbable blast media (RBM) and sandblasted and acid-etched (SLA) surface implants were found to be safely preferable with proper case selection [8]. Furthermore, UMUT-SLA and tricalcium phosphate/hydroxyapatite (TCP/HA) surfaces improved the BIC value, and induced accumulated bone density during the osseointegration period (six weeks) [9,10]. Moreover, modifying the surface of the titanium implant with ultrafine-grain titanium (UFG-59-Ti) results in superior mechanical properties and maintained cytocompatibility and osseointegration potential [11].

Successful implantation with proper osteointegration is mediated through oral tissue enhancement and prevention of bacterial infections. Therefore, several studies have recommended the use of bioactive stimuli. Serum proteins, melatonin, and other bone stimulators, adsorbed at the implant surface, were found to mediate cell adhesion through integrin receptors. Meanwhile, the release of molecules post-surgical trauma promotes platelet aggregation and the coagulation cascade, which induces fibrin clot formation that could act as a scaffold for the migration of undifferentiated cells and osteoblastic precursors [12,13,14]. Graphene-chitosan hybrid dental implants were reported to promote osteoblast proliferation while reducing biofilm formation and bacterial activity [15]. Similar observations were also reported for titanium implants treated with zinc-containing tricalcium phosphate (Zn-TCP), poly-ε-caprolactone/titania (PCL/TiO_2_), or plasma-sprayed cerium oxide (CeO_2_) [16,17,18,19]. Other titanium implant-coating materials, such as nano-Ag and poly-ethylene oxide (PEO), possess notable antibacterial properties [20,21].

Wettability is one of the factors that is able to influence the earlier stages of osseointegration, in particular, the adsorption of proteins on the implant surface, quantity of bound proteins, binding strength, conformation, and consequently the adhesion of soft and hard tissue cells on preconditioned surfaces [22,23]. In contact with blood cells, hydrophilic surfaces promote protein absorption and expose adhesion sequences, which bind integrins onto the cell membrane. Counter wise, hydrophobic surfaces denature proteins, reduce cell binding sites, and provoke cell adhesion [24]. Therefore, hydrophilic surfaces, compared to hydrophobic ones, improve adhesion and fibroblast proliferation [25], enhance osteoblast maturation, and mediate the expression of differentiation markers [22]. Bang et al. reported that SLA and modified-SLA surfaces promote osteogenic and anti-osteoclastic effects, sustain marginal bone maintenance, and increase the values of resonance frequency analysis (RFA), implant stability quotients (ISQs), and BIC percentage [26,27]. In vivo studies are in alignment with the described in vitro observations, with notable elevation BIC values and a reduction of torque values using implants treated with hydrophilic surfaces [28,29,30].

Applying bioactive molecules, such as cholecalciferol/vitamin D3 or silicon-doped ti-87 tanium dioxide (Si-doped TiO_2_) nanotubes, onto titanium dental implants improves the corrosion resistance against aggressive oral cavity fluids and facilitates osteoblastic bone cell adhesion [31,32,33]. Furthermore, titanium implants treated with liquid plasma rich in growth factors (PRGF) mediated efficient osseointegration, bone regeneration, and higher BIC values when compared to the control untreated implant [34,35,36]. The combination of calcium ion surfaces with PRGF enhanced bone regeneration within two weeks of implantation [37,38].

The aim of this study was to compare the crestal bone loss, bone formation, and soft tissue width at 8- and 12-week assessment intervals for two different SLA implant bioactive surfaces: bone level tapered (BLT) implants with SLA active (Straumann, Basel, Switzerland, control group) and with BLT-SLA-BBL (test group) (BBL, patent # EP353211, US 16/344,322).

## 2. Material and Methods

### 2.1. Statement of Compliance and Declaration of the Research Ethics

The local Experimental Animal Research Ethics Committee (Comite Etico de Experimentacion Animal, CEEA) at The University of Murcia, Murcia, Spain, approved the present study protocol using dog animal model article number 34- RD 53/2013, project number A1320141102. In addition, the research project was conducted and adhered to the guidelines of the Animal Research: Reporting of In vivo Experiments (ARRIVE). The authors declare adherence to the proper institutional and national guidelines related to the care and use of the animals in this study including ethics initiative, design, analysis, and reporting of research using animals. Furthermore, the timeline of the present study is described in Figure 1A. During the entire treatment period, the animals’ behavior, posture, reactivity, and appearance were monitored by a professional veterinary doctor.

### 2.2. Bone Bioactive Liquid and Implant Design

Here, we describe a new bone bioactive liquid (BBL, patent # EP353211, US 16/344,322) comprising a saline solution (PBS) containing calcium chloride (CaCl_2_) and magnesium chloride (MgCl_2_-6H_2_O) with a net negative charge that creates an ideal condition for cellular attraction to the trauma area at the bone–implant contact. Apart from that, BBL considerably increases the density of hydroxyl groups on the wound implant surface and improves their hydration significantly. BBL induces surface hydrophilicity, and allows its’ active ionic interaction with blood plasma and bone progenitor cells on the implant’s surface. As a result, it leads to coordination and communication between the cells and to better contact between all tissues in the wound for “new tissue formation”.

A total of 40 Straumann BLT implants with SLA alone or SLA active (Straumann, Basel, Switzerland) were utilized in this study. The implants had an endosteal diameter of 3.3 mm, length of 10 mm, and were designed to be placed at bone level in healed sites as described by the manufacturer. Implants were divided into two groups (20 each). The first group of SLA implants were treated with BBL for 24 h (test group) and the second control group were SLA active implants.

### 2.3. Tooth Extraction, Surgical Procedure, and Animal Care

Four American foxhound adult male dogs were used in the study, in accordance with the permit from Experimental Animal Research Ethics Committee, University of Murcia, Murcia, Spain. Animals were anesthetized by femoral quadriceps intramuscular injections with a cocktail containing acepromazine (0.12–0.25 mg/kg), buprenorphine (0.01 mg/kg), and medetomidine (35 mg/kg). Animals were then transferred to the operating room. General anesthesia was sustained by a continuous propofol (0.4 mg/kg/min) infusion into the cephalic vein though an intravenous catheter insertion. In addition, conventional local dental anesthesia, 4% Articaine-hydrochloride and 0.001% epinephrine, was injected at the intraoral surgical sites as described by [39]. The operation was performed by one dental surgeon and under the supervision of a veterinary specialist. 

The mandibular premolars P2, P3, P4, and the molar M1 were extracted from both sides carefully via a minimally invasive surgical approach. Teeth were sectioned in a buccolingual direction using a tungsten-carbide bur to facilitate individual root’s extraction without damaging the remaining alveolar bone walls and the alveolus was left to heal as previously described [40]. After a healing period of 12 weeks, for implant placement, full-thickness mucoperiosteal flaps were raised in the hole mandible (Figure 1B).

Each dog received 5 implants inserted at the healed sites of each hemi-mandible using a motorized hand piece and were subsequently equipped with healing abutments, which were placed 0.5 mm below the occlusal contact with the corresponding antagonist tooth. The implants were arranged into two experimental groups: control group: BLT-SLA active and test group: BLT-SLA +BBL (Figure 1C,D). No graft materials were placed between implants and bony plates. Mucoperiosteal flaps were stitched with Silk 4-0 non-absorbable sutures (Lorca Marin, Lorca, Murcia, Spain). To prevent infections, antibiotics were administered for seven days, 500 mg Amoxicillin, and 600 mg Ibuprofen, two and three times a day, respectively. For a period of seven days, animals were fed a soft diet. To prevent plaque formation, animal received seawater-based mouth wash rinse with SEA4-Encias (Blue Sea Laboratories, Alicante, Spain). Wound healing was monitored daily to exclude possible clinical complications. Sutures were removed two weeks post-surgery.

### 2.4. Termination

At a two-time interval, 8 and 12 weeks post-implantation, two dogs were sacrificed by carotid arteries perfusion with fixative solution containing Pentothal Natrium (Abbot Laboratories, Madrid, Spain), 5% glutaraldehyde, and 5% formaldehyde. The hemi-mandibles were dissected, washed in phosphate-buffered saline (PBA), fixed in 10% formalin (Sigma-Aldrich, Germany), and processed at BioTecnos Laboratories (Santa Maria, Brazil).

### 2.5. Histologic Processing

Hemi-mandible block sections were dehydrated in a series of ascending concertation’s of graded ethanol up to 100%, and then infiltrated and embedded using Technovit-7200 VLC system (Kulzer Technique, Wehrheim, Germany) in accordance with the manufacturer’s protocol. After chemical polymerization, samples were cut in buccolingual direction to sections at 2.5 mm and then refined to 35–50 µm in thickness, using IsoMet-1000 high precision diamond disk (Buehler, Illinois, IL, USA). Slides were preserved for future use. For each implant, two slides were incubated for 1 h. at 25 °C in Picrosirius Red Staining (PSR) solution (Polysciences, Warrington, PA, USA).

### 2.6. Histomorphometric Analysis and Examination

Histomorphometric analysis were performed using Image J software, version 1.38e. Measurements were performed at the region of interest (ROI), defined as the peri-implant, located between 3 and 6 mm below the implant shoulder at the central mesiodistal sections. Measurements were performed for the inter-thread bone area percentage (ITBA%), the bone to implant contact (BIC%), and the crestal bone loss (CBL) for the lingual and buccal bone areas (Figure 1E). Linear measurements were made at the buccal (BBC) and lingual (LBC) bone crest starting from the implant shoulder (IS) to the first point of BIC contact to BBC (IS-BC) or to the LBC (LC-IS). The BIC percentage of native bone was measured throughout the implant surface as described in [41] (Figure 1E). The gingival tissue restorations, in the vertical and lateral direction, were measured at the implant neck throughout the thickness (points A-B) and at the abutment to the bone crest (points C-D) (Figure 1E). Measurements were performed by an expert histology examiner (JLCG). Metric evaluation of the predetermined parameters was carried out using a light microscope (Nikon, Tokyo, Japan) connected to a high-resolution video camera (3CCD, JVC KY-F55B, Yokohama, Japan). After digitizing the phase of each specimen under the light microscope, all the predetermined distances were measured on images using the program Image Tool version 5.02 for Microsoft Windows (UT Health Science Center School of Dentistry, San Antonio, TX, USA).

### 2.7. Statistical Analysis

To test for differences between the paired data within animals, Wilcoxon signed rank test was used. The associations between the measured outcomes and test were evaluated using mixed linear regression models, considering the factor position of the test and the random animal effect. To evaluate if the measured outcomes correlated with each other at the time points, the Pearson correlation coefficients were calculated. Means and standard deviations of the crestal bone height and tissue thickness were calculated for all groups. The Mann–Whitney normality test was applied for all measurements. After applying the Levene test for equality of variances, one-way ANOVA tests were used to identify significant differences in the IS-LC and BC-IS parameters among groups at the buccal and lingual aspects. The Student–Newman–Keuls test was applied to make pairwise comparisons. All analyses were performed with specialist software (MedCalc Statistical Software version 15.8; MedCalc Software bvba, Ostend, Belgium). The statistical significance was set at 5% (*p* < 0.05).

## 3. Results

In general, no implants were lost during the study, but notably, surgical sites healed differentially and inconsistently. All animals presented an appropriate healing during the first two week after the surgical procedure, with no pathogen infections or tissue inflammation. All implants presented osseointegration after the 8- and 12-week period and were available for histological analysis. Accordingly, data were collected from all implants with no exclusions.

In all studied groups, at week 8 of implant assessment, direct contact was observed between implants and living bone in the absence of soft tissues. At implantation week 8, histological assessment of the BLT-SLA active implants revealed a notable integration and bone remodeling around the implant, slight resorption in the buccal wall, and soft tissue stability at both sides surrounding the implant abutment (Figure 2A). On the other hand, the BLT-SLA pretreated with BBL displayed an improved soft tissue and bone remodeling around the implant surface at both the buccal and lingual bone crests and above the implant neck (Figure 2B). 

At 12 weeks post implantation, the BLT-SLA active implants presented with pronounced stable soft and hard tissue generation at both the buccal and lingual crestal bone, which was also sustained above the implant neck (Figure 2C). At the buccal wall, the SLA implants pretreated with BBL presented with bone enrichment and soft tissue eradication; in contrast, bone that kept the implant neck and soft tissues was enhanced at the lingual wall crest (Figure 2D).

### 3.1. Bone Integration and Density

Bone integration was measured by BIC progression, which showed a prominent improvement (>60%) over the course of the healing range between. Statistically significant elevated values in BIC were observed for the BLT-SLA + BBL test group, relative to that of the control BLT-SLA active group at both assessment week 8 and week 12 post implantation (Figure 3). 

Bone density was measured as ITBA percentage values. Like BIC values, we observed a notable improvement in ITBA values during the healing process. Interestingly, the test group implants showed a significant increase, 1.5 to 1.7-fold, in the ITBA bone regeneration values relative to the control BLT-SLA active group (Figure 3 and elucidated by representative Figure 4A,B, respectively).

### 3.2. Crestal Bones Loss and Tissue Thickness

Next, we evaluated the crestal bone regeneration and loss in both the control and test groups. At assessment week 8, the control group possessed new bone formation at the lingual but not the buccal bone crest, which showed slight bone reportion. However, the soft tissues were rigid at both crests (Figure 5A,B, respectively). At this interval, the BLT-SLA pre-treated BBL, the test group, showed active crestal bone regeneration exceeding the level of the implant neck, a process that was associated with enrichment of soft tissues at both the lingual and buccal aspects (Figure 5C,D, respectively). 

At assessment week 12 post-implant, an increase in bone remodeling and maturation was observed at both the lingua and buccal crest of the control BLT-SLA active group flanking the implant neck (Figure 5E,F, respectively). At this group, the soft tissue showed no signs of inflammation and remained stable above the ridge with adequate thickness. On the other hand, ameliorated bone maturation was detected at the both buccal and lingual walls of the test SLA + BBL group (Figure 5G,H). Mature bone extended along the ridge to the implant neck and the gingival tissue was stable with a semi-normal thickness. In general, greater bone preservation was noticed at the crests of BLT-SLA pretreated with BBL compared to control BLT-SLA active implants at both assessment periods.

Crestal bone loss was determined by measurements of the distance between the implant collar top (line A) and the point of contact with bone (line B) as indicated in Figure 1E. Interestingly, both the buccal and lingual dimensions showed statistically significant differences at weeks 8 and 12 post-implantation between the test and control groups. As shown in Figure 6, the A-B distance evaluated in the test BLT-SLA + BBL group was significantly lower in both buccal and lingual at both assessment intervals, relative to that of the control BLT-SLA active group. Tissue thickness was also measured by utilizing the distance between the implant collar top (line C) and the more external portion of the tissues (line D) (Figure 1E). The crestal bone height and tissue thickness were slightly significantly higher for the test group at both the buccal and lingual sites at assessment week 8. At week 12, comparable values to that of week 8 between the test and control groups were observed (Figure 6).

## 4. Discussion

The main purpose of implant surface modification is to modulate the host tissue in favor of osseointegration. In this animal model study, we compared the osseointegration performance of BBL, a bone liquid developed in our laboratories, and the commercially available SLA active titanium bioactive surfaces at 8 and 12 weeks post-implantation. SLA active titanium is a notable implant brand with an efficient bioreactivity and healing characteristics [42,43]. Implants were evaluated histologically by means of BIC, ITBA, crestal bone loss, tissue thickness, and soft tissue regeneration, requisite parameters for new bone regeneration and implant stability and indicative of successful osteogenesis [44].

BBL is a saline solution mainly composed CaCl_2_ and MgCl_2_-6H_2_O and adjusted with a negatively charged electrolytes that is suitable for enhancing biological reactivity at the implant site. Interestingly, histomorphometric analysis revealed that the BIC and ITBA values of BLT-SLA pre-treated with BBL are significantly higher than that of the SLA stored in sodium chloride solution BLT-SLA active implants, indicating an improvement in fibrin network organization, new bone formation, and implant stability. Moreover, the elevated BIC values in the test group are an indication of pronounced hydrophilic properties, a finding that is consistent with previous studies [30,45]. Due to the liquid nature of BBL and SLA-active, the data also indicates an improvement in the surface wettability and roughness of the treated titanium implants.

Animal studies indicate that the best results are obtained with implants/bioactive with combined mechanical and chemical treatment methods, thus obtaining better bone-to-implant contact for implants and improving osseointegration mediated by bioactive surfaces, relative to untreated controls (best described in the recent systematic review [46]). In accordance, the current study evaluated BLT-SLA pretreated with the bioactive BBL. BLT-SLA is a titanium implant that is sandblasted with long-grit corundum and refined with acid treatments, a process that provides sufficient mechanical roughness. BLT-SLA pretreatment with the bioactive BBL provided the second condition for implant improvements, as observed with the notable minimization in crestal bone loss and enhancement of soft and hard tissue generation and thickness, particularly at week 12 post-implantation. A similar bioreactivity was observed for the control BLT-SLA active implants but albeit at a significant lower competency. These results suggest that the salt combination and negative ionic charge of BBL were sufficient to induce tissue remodeling, osteoblast activity, and/or prospective recruitment of circulating cells rich with growth-inducing factors.

Several other studies reported improvements on SLA surfaces using bioactive materials, indicating that the SLA surface roughness is ideal for implantation, yet there is still room to improve its bioreactivity by modifying the surface with ingredients that are sufficient to enhance osseointegration and bone healing. In an earlier animal study, Buser et al. observed an enhancement in bone apposition during the early stages of bone generation using a SLA titanium surface treated with isotonic NaCl solution (SLA active), relative to the control untreated implants. Similarly, SLA pretreated with hydroxide ions (conSF) showed an increase in mineralized bone-implant contact after a short healing process, relative to untreated controls [45]. In a human randomized controlled clinical trial, Oates et al. concluded that chemically modified SLA active surface implants may alter biologic events during the osseointegration process, which in turn enhances the healing process, and therefore could lead to adjustments in clinical loading protocols for dental implant therapy [47]. In a comparative study, Lee et al. evaluated the effectiveness of bone healing and remodeling of three commercially available implants: a hydrophobic SLA-IS-III active, a hydrophilic SLA surface implant with HA nanocoating (IS-III Bioactive), and a hydrophilic SLA active [43]. These authors observed higher and a faster new bone generation associated with the use of hydrophilic surface implants in comparison to hydrophobic surface implants, whereas the use of HA-coated surface implants facilitated osteoblast activity [43]. Taken together, these observations reveal the importance of the implant surface’s wettability, which most likely facilitates bone healing dynamics around implants and may be attributed to the improvement of early osseointegration.

The use of liquid bioactive material improves implant wettability and influences the early stages of osteointegration. However, the reactivity of liquid bioactive surfaces is dependent on their hydrophilic competency, salt composition and concertation, electrodynamic charge, lack of solid components, and proteins, factors that may influence viscosity, stiffness, and roughness [29,48,49]. Anitua et al.’s innovative work using titanium implants pre-treated with liquid PRGF proved that the implants accelerate bone regeneration, reduce post-extraction defects, and thus, shorten the time between tooth extraction and implant insertion [34,35,36,38,50,51]. Liquid PRGF is rich in anti-inflammatory and growth factors that modulate the immune response, enhance circulating cell recruitment, and mediate cellular proliferation and differentiation at the implant healing site [52]. Utilizing a similar rational, Scarano et al. developed autologous platelet liquid (APL) for implant site irrigation, implant immersion, or to be mixed with biomaterials during regenerative procedures [53]. 

Several studies have reported that Ca^2+^- and Mg^2+^-containing bioactive implants, such as hydroxyapatite (HA), improve material porosity and crystallite and upregulate the expression of genes associated with bone formation [54,55]. Ratanyake et al. observed a substantial improvement in the bioactivity of stoichiometric HA pot treated with cationic and anionic sublattices [56]. Similar observations were also reported for other HA modifications including cations, such as Sr^2+^, Pb^2+^, or Mg^2+^, and anions, such as F^−^ or Cl^−^ (best reviewed in [57]). Interestingly, there modifications are sufficient to provide effective barriers against metal corrosion progress and could possess biocompatibility and osteoconductive properties [58].

## 5. Conclusions

This study examined and compared the biological parameters of a novel bioactive surface implant relative to that of a commercially available surface material that is widely used in dental clinics and known for its favorable characteristics post-transplantation. Our data indicate that implants pre-treated with BBL possess comparable and superior bioreactivity characteristics when compared to SLA active. Within the limitations of this study, our data revealed that BLT-SLA pre-treated with BBL reduced the titanium surface roughness and improved wettability, sufficient characteristics to promote healing, eradicate inflammation, and promote bone and gingiva regeneration at the implant site.

The main limitation of this study is the sample size, as due to ethical restrictions, it is relatively small. More studies are required with an increase in sample size to confirm the current research outcome and BBL application with a variety of rough titanium implants from different brands. Hopefully, randomized controlled clinical trials will also provide evidence of the efficacy of BBL implant treatment in bone regeneration, inflammation and pathologic resistance, and implant stability.

## Figures and Tables

**Figure 1 materials-14-06217-f001:**
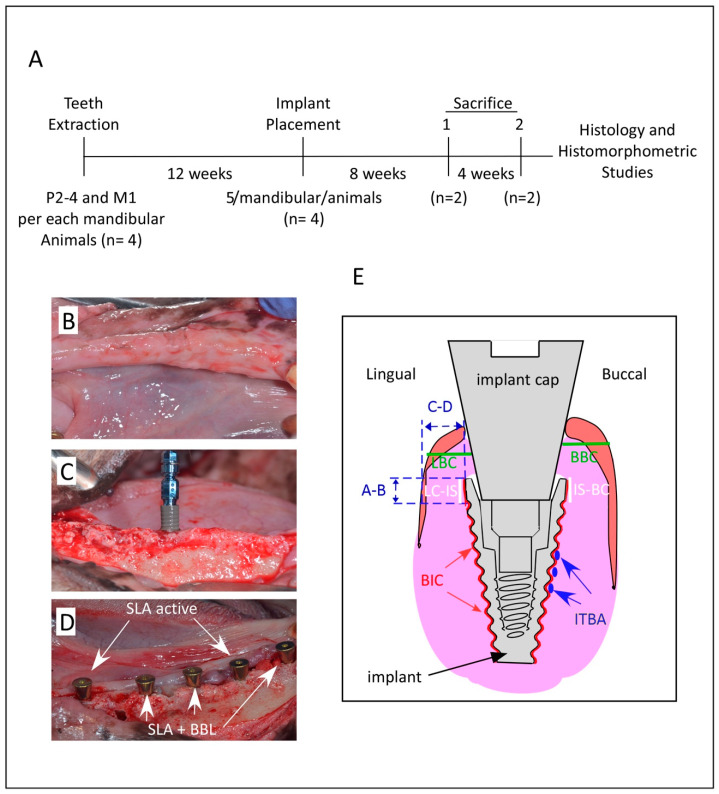
Experimental Procedure. (**A**) Timeline of the current study. Representative images for (**B**) healed bone after 12 weeks post extraction; (**C**) Straumann BLT-SLA implant with BBL liquid; and (**D**) five implants with healing abutments in place with the distribution of BLT-SLA implants with BBL pre-treatment (test group) and BLT-SLA active (control group). (**E**) Illustration of histomorphometric parameter measurements. Total bone-to-implant contact (BIC) was measured around the total endosseous circumference of the implant as the percentage the implant surface in contact with bone over the total implant surface (red line). Inter-thread bone area percentage (ITBA) is illustrated in the yellow insert. Buccal (BBC) and lingual (LBC) bone crest measurements from the implant shoulder (IS) to BIC contact to BBC (IS-BC) and LBC (LC-IS), respectively. Crestal bone loss measurements vertically, the distance between the implant collar and the first crestal bone contact = A-B is bone height. Tissue thickness measurements horizontally, the distance between the implant collar to the external portion of the tissues = C-D is tissue thickness.

**Figure 2 materials-14-06217-f002:**
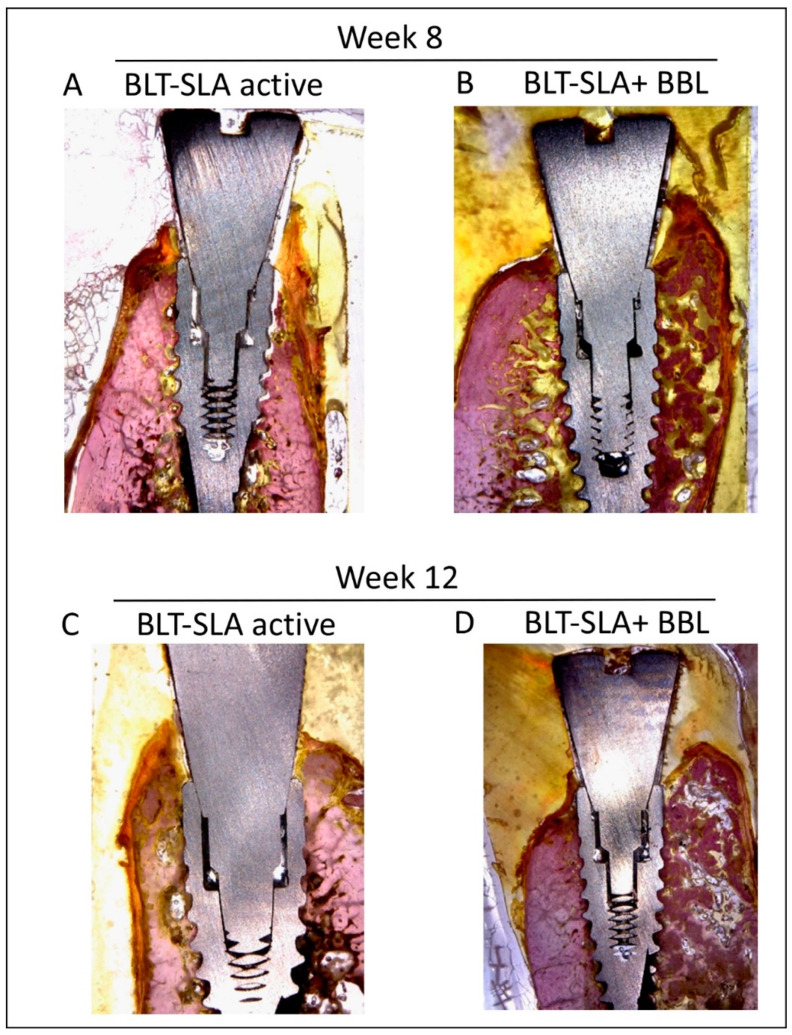
Representative image Illustrating the implants. (**A**) Week 8, BLT-SLA active implant showed moderate bone remodeling with slight resorption at the buccal wall. (**B**) Week 8, BLT-SLA + BBL liquid represented enriched bone remodeling and increased bone crest above the neck of the implant at both the buccal and lingual wall crest. (**C**) Week 12, BLT-SLA active showed stable generation of soft and hard tissues flanking the buccal and lingual crests. (**D**) Week 12, BLT-SLA + BBL liquid showed an enrichment of bone at the buccal wall, whereas soft tissue enrichment was observed at the lingual wall. All images were taken at magnification 12.5×.

**Figure 3 materials-14-06217-f003:**
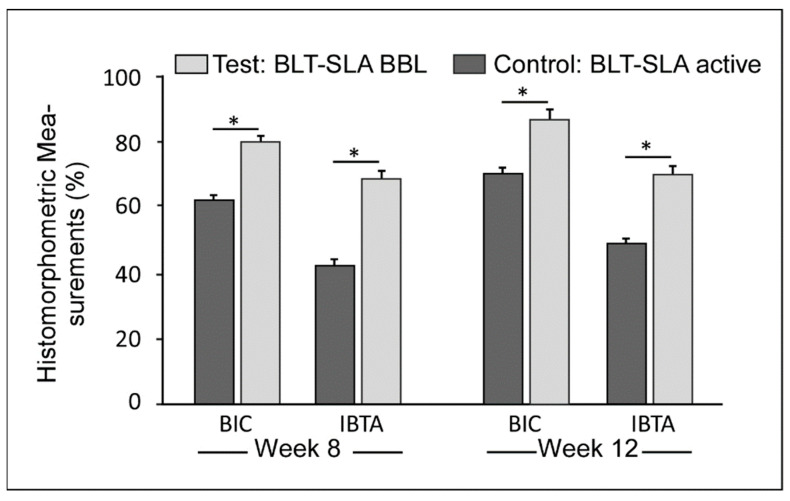
The bone-to-implant contact (BIC) and inter-thread bone area (ITBA) measurements at the 8- and 12-week interval post-implantation. (* *p* < 0.05).

**Figure 4 materials-14-06217-f004:**
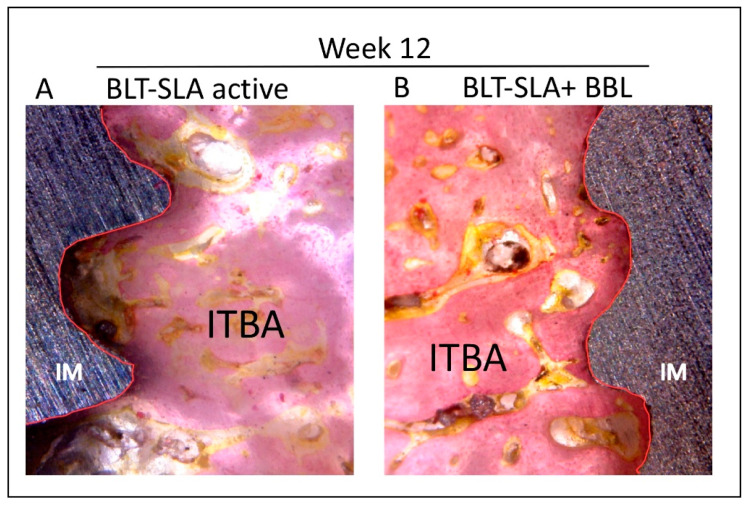
Representative image illustrating the inter-thread bone area (ITBA) generated at week 12 post-implantation. (**A**) Control group, BLT-SLA active implant. (**B**) Test group, BLT-SLA pre-treated with BBL. At the week-12 assessment, new ITBA is generated in the test group, but SLA active showed sustained bone remodeling around the threads’ pitch and valley. IM = implant. Magnification 24×.

**Figure 5 materials-14-06217-f005:**
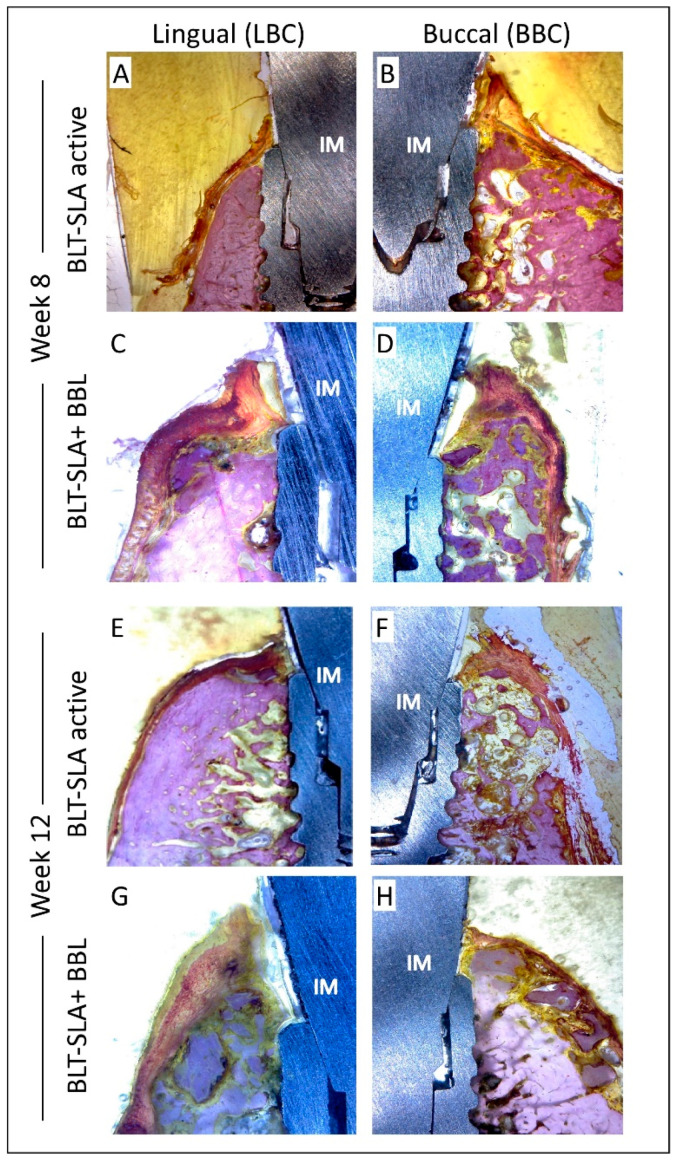
Representative image illustrating the crestal bone loss at week 8 (**A**–**D**) and 12 (**E**–**H**) post-implantation assessment. Control group, BLT-SLA active implant A and E lingual (LBC) and B and F buccal (BBC) bone crest. Test group, BLT-SLA pre-treated with BBL. C and G lingual (LBC) and D and H buccal (BBC) bone crest. Pronounced bone and soft tissue regeneration was observed in the test group at week 8. Whereas, at week 12, significant bone maturation and soft tissue amelioration was observed in the test group in comparison to the control group. Magnification 16×.

**Figure 6 materials-14-06217-f006:**
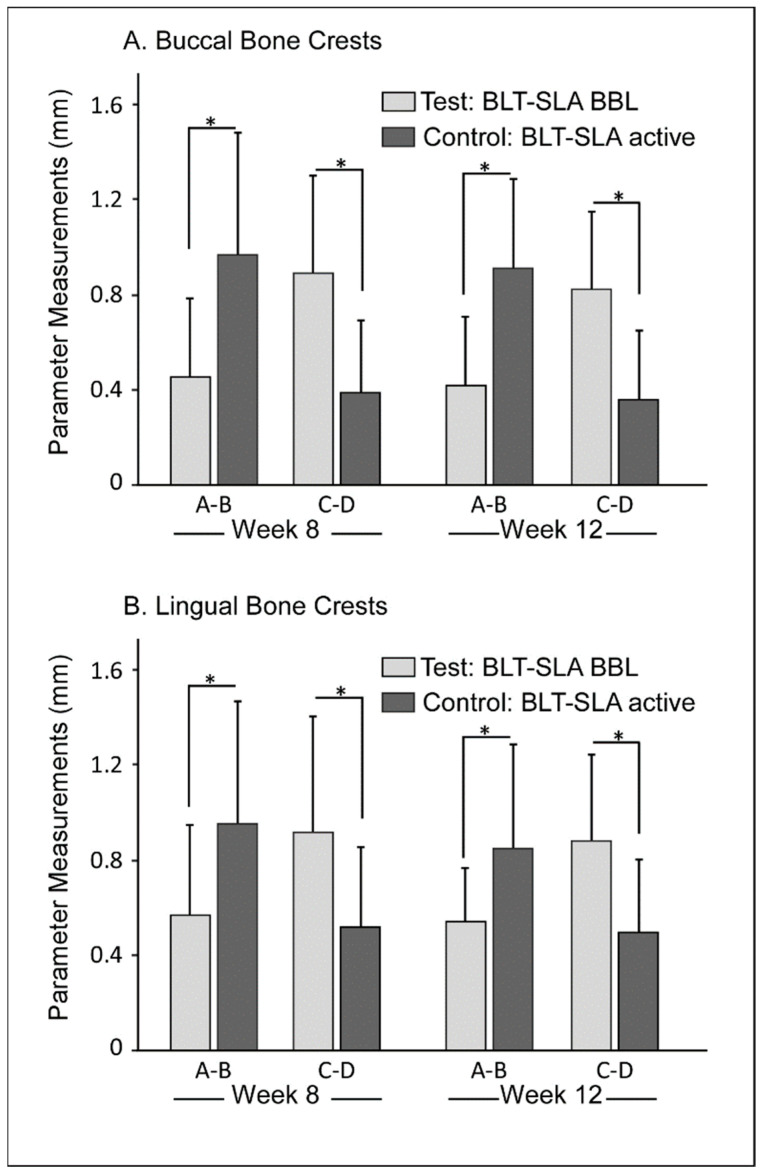
Crestal bone loss and tissue thickness (mm) at 8 and 12 weeks post-implantation. Buccal and lingual bone crests within each parameter (A–D) represent slightly statistically significant differences for pairwise comparisons (* *p* < 0.05).

## Data Availability

Data available on request due to restrictions eg privacy or ethical. The data presented in this study are available on request from the corresponding author.

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
