# Peer review of "Histologic and Histomorphometric Evaluation of a New Bioactive Liquid BBL on Implant Surface: A Preclinical Study in Foxhound Dogs"

_materials, 2021, doi:10.3390/ma14206217_

Round 1
Reviewer 1 Report
I believe that the work submitted for review is suitable for publication. The background is sufficiently explained and the study design is clear. The language used is fluent; in the text there are only a few minor typos to be corrected (for instance, it's not clear the sentence in page 2 line 52; In Figure 1, line 126, there is an excess parenthesis).
Author Response
We would like to thank the reviewer for his/her support to improve the manuscript. We corrected the sentence Page 2 Line 52. Line 126 was also corrected.

Reviewer 2 Report
Although this is an interesting study, there are still many areas to improve and verify the results, especially the lack of major establishment and analysis of biologically active treatment.
Specific suggestions:
Abstract
It is recommended to give reliable statistical values and comparison results. I suggest modifying the expressions in the results and conclusions of abstract.
Introduction-
2nd paragraph as a revision example-
“Bone-to-implant contact (BIC). Several approaches have been applied to augment the 52 BIC.” For example, Korean implants have been UV-irradiated prior to implantation in 53 dogs, with no statistically significant differences in the mean insertion torque or BIC val-54 ues in all groups “at four and 12 weeks” [7] . Consequently, resorbable blast media (RBM) 55 and sandblasted and acid etched (SLA) surface implants were found to be safely prefera-56 ble with proper case selection [8]. “UMUT-SLA and tricalcium phosphate/hydroxyapatite” 57 (TCP/HA surfaces improved BIC value, and induced accumulated bone density during 58 the osseointegration period (six weeks) [9; 10]. - The text in many places is informal and needs to be corrected.
Moreover, ultrafine-grain titanium (UFG‐59 Ti) exhibits superior mechanical properties while maintaining cytocompatibility and os-60 seointegration potential [11]. - This means that the surface of the titanium implant modified with ultra-fine crystal titanium and the reference should be more specific.
The use of graphene-chitosan hybrid dental im-66 plant surface with enhanced antibacterial and cell proliferation properties [15], similarly, 67 electrospinning of poly-ε-caprolactone/titania (PCL/TiO2), poly-ethylene oxide (PEO), 68 nano-Ag, FZn-CaP, and Plasma-Sprayed cerium oxide (CeO2) coatings were successfully 69 congregated onto the surface of titanium implants, and possessed surface property alter-70 ations including wettability and roughness [16; 17; 18].- Same as the above question, the cited references are not specific
Silicon-doped ti-87 tanium dioxide (Si-doped TiO2)- needs to be corrected and checked throught the manuscript.
Last paragraph- This paragraph suggests moving to materials and methods. The author is strongly recommended to provide detailed informations on the surface modification process, conditions and results, including SLA and SLA-BBL due to it is the major finding.
Materials and Methods
It is recommended to provide real images and microstructures of the implant surface, at least citing the above reference materials.
Results- Figures 2-4 - Poor image resolution makes it difficult to verify the results. Can the authors improve it?
Author Response
Reviewer 2
Comments and Suggestions for Authors
Although this is an interesting study, there are still many areas to improve and verify the results, especially the lack of major establishment and analysis of biologically active treatment.
We would like to thank the reviewer for his/her interests in our study, and the kind support to improve the manuscript.
Specific suggestions:
Abstract
It is recommended to give reliable statistical values and comparison results. I suggest modifying the expressions in the results and conclusions of abstract.
We agree with the reviewer, this section has improved the manuscript. We added the suggested statistical comparison between the test and control group. Lines 32-41.
Introduction-
2nd paragraph as a revision example-
“Bone-to-implant contact (BIC). Several approaches have been applied to augment the 52 BIC.” For example, Korean implants have been UV-irradiated prior to implantation in 53 dogs, with no statistically significant differences in the mean insertion torque or BIC val-54 ues in all groups “at four and 12 weeks” [7] . Consequently, resorbable blast media (RBM) 55 and sandblasted and acid etched (SLA) surface implants were found to be safely prefera-56 ble with proper case selection [8]. “UMUT-SLA and tricalcium phosphate/hydroxyapatite” 57 (TCP/HA surfaces improved BIC value, and induced accumulated bone density during 58 the osseointegration period (six weeks) [9; 10]. - The text in many places is informal and needs to be corrected.
We agree with the reviewer, this paragraph has been improved (Lines 57-66). We are planning to send the manuscript for English Language editing after acceptance to reduce costs.
Moreover, ultrafine-grain titanium (UFG‐59 Ti) exhibits superior mechanical properties while maintaining cytocompatibility and os-60 seointegration potential [11]. - This means that the surface of the titanium implant modified with ultra-fine crystal titanium and the reference should be more specific.
We agree with the reviewer, this paragraph has been improved in accordance with your suggestions and reference 11. (lines 67-68)
The use of graphene-chitosan hybrid dental im-66 plant surface with enhanced antibacterial and cell proliferation properties [15], similarly, 67 electrospinning of poly-ε-caprolactone/titania (PCL/TiO2), poly-ethylene oxide (PEO), 68 nano-Ag, FZn-CaP, and Plasma-Sprayed cerium oxide (CeO2) coatings were successfully 69 congregated onto the surface of titanium implants, and possessed surface property alter-70 ations including wettability and roughness [16; 17; 18].- Same as the above question, the cited references are not specific
We agree with the reviewer, this paragraph has been improved in accordance with your suggestions and references were specified and corrected. Lines 76-86. We also induced 4 extra references to clarify the idea.
Silicon-doped ti-87 tanium dioxide (Si-doped TiO2)- needs to be corrected and checked throught the manuscript.
We agree with the reviewer, corrections were done.
Last paragraph- This paragraph suggests moving to materials and methods. The author is strongly recommended to provide detailed informations on the surface modification process, conditions and results, including SLA and SLA-BBL due to it is the major finding.
We agree with the reviewer, this section has improved the manuscript and already moved to M&M also we added the suggested detailed information on the surface modification process (lines. 111-115 and 139-148)
Materials and Methods
It is recommended to provide real images and microstructures of the implant surface, at least citing the above reference materials.
We appreciate the reviewer recommendation; we are finishing another paper which will be about the full chemical and physical characteristics of our new BBL treated surface, so we prefer to keep the microstructure images for the second paper, In the supplemental figure we included the original images.
Results- Figures 2-4 - Poor image resolution makes it difficult to verify the results. Can the authors improve it?
We thank the reviewer for the suggestion. we have improved all the images of the requested figures also we are providing the original Tif files as a separate image in the supplementary materials.

Reviewer 3 Report
The article has a lot of methodological shortcomings, so I recommend major revision before considering it for publication. Basing the entire publication on one type of method certainly does not raise the rank of the article. The authors should at least try to use the simplest staining methods to analyze, for example, tissue loss.
M&M: Why did the authors not lead the control in the form of untreated BBL implants?
M&M: Please explain, why 8 and 12 weeks were chosen for sacrifying the animals?
M&M: 3.2. Crestal Bones Loss and Tissue Thickness. In my opinion the more valuable method for measuring the crestal bones loss would be the X-ray or CT scanning. The histological measurements are very subjective. So I strongly recommend, if it is possible, to determine bone loss with one of mentioned techniques.
Figures: 2 and 4 – please indicate the names of the groups in the Figure to be easier to follow, not only in the legends.
Please replace tables with values with bar charts, will be much clearer and it will be easier to spot the difference.
Table 2: Are the authors sure of the differences presented between the groups? The numerical values are not much different from each other, so the significant difference between them is quite surprising.
The discussion is short and does not compare the authors' results with others. I recommend expanding and discussing the results more strongly.
Author Response
Reviewer 3
Comments and Suggestions for Authors
The article has a lot of methodological shortcomings, so I recommend major revision before considering it for publication. Basing the entire publication on one type of method certainly does not raise the rank of the article. The authors should at least try to use the simplest staining methods to analyze, for example, tissue loss.
M&M: Why did the authors not lead the control in the form of untreated BBL implants?
We thank the reviewer for the suggestion. The study was a comparison between the bioactivity of the liquid coated materials, the BLT-SLA treated with BBL under-study, and the commercially available BLT-SLA active, soaked in NaCl. SLA active was reported to be superior to untreated SLA as described in the publication doi: 10.5051/jpis.2019.49.1.25. Further, the main differences is that BBL solution is composed CaCl2 and MgCl2-6H2O and adjusted with a negatively charge electrolytes.
M&M: Please explain, why 8 and 12 weeks were chosen for sacrifying the animals?
We thank the reviewer for the qualification. The 8- and 12- weeks’ intervals to scarify the animal after implantation, are common practices in the field, and in order to be in alignment with other studies, we chosen these intervals.
M&M: 3.2. Crestal Bones Loss and Tissue Thickness. In my opinion the more valuable method for measuring the crestal bones loss would be the X-ray or CT scanning. The histological measurements are very subjective. So I strongly recommend, if it is possible, to determine bone loss with one of mentioned techniques.
We thank the reviewer for the suggestion; however, these techniques are not available at our facilities and we were plaining to do in our initial proposal, however, we find it hard to achieve. Nevertheless, the histological measurements approach has been used by several scientists and has been described in several other publication uses
Figures: 2 and 4 – please indicate the names of the groups in the Figure to be easier to follow, not only in the legends.
We thank the reviewer for the suggestion; Figures were modified accordingly.
Please replace tables with values with bar charts, will be much clearer and it will be easier to spot the difference.
We thank the reviewer for the suggestion; Tables 1 and 2 were replaced by Figures 3 and 6.
Table 2: Are the authors sure of the differences presented between the groups? The numerical values are not much different from each other, so the significant difference between them is quite surprising.
We thank the reviewer for the excellent observation. We revised the data and there was a mistake in the data presentation. The idea of preparing tables was not good as the figures! We have noticed that in Table 2 the data were switched due to personal error. The values for the A-B and C-D were switched by that of Test versus Control (the Table 2 should read as below). In the article, the table 2 was replaced by Figure 6. Accordingly, we have corrected the error. Further, in human and animal studies we and other has noticed a statistically significant differences between treatments, although the SD bars could overlap. We really like to thanks to the reviewer’s sharp observation that prevented a mistake.
|
Tissue |
Parameters Mean (mm ± SD) |
Week 8 |
Week 12 |
||
|
A-B |
C-D |
A-B |
C-D |
||
|
Buccal |
Test |
0.47* ± 0.32 |
0.89* ± 0.41 |
0.41* ± 0.29 |
0.82* ± 0.33 |
|
Control |
0.98 ± 0.51 |
0.39* ± 0.3 |
0.91 ± 0.38 |
0.36* ± 0.29 |
|
|
Lingual |
Test |
0.57* ± 0.38 |
0.92 ± 0.48 |
0.54* ± 0.23 |
0.88 ± 0.36 |
|
Control |
0.96 ± 0.51 |
0.52* ± 0.33 |
0.87 ± 0.41 |
0.49* ± 0.31 |
|
The discussion is short and does not compare the authors' results with others. I recommend expanding and discussing the results more strongly.
We thank for the reviewer for his/her great input to improve out manuscript. We have updated the discussion as recommended, by adding more comparisons and references to other similar studies. Lines 339-374 and 390-395).

Round 2
Reviewer 2 Report
The author continuously revised the revised edition to a large extent, therefore, it could be acceptable.
Reviewer 3 Report
The authors followed the suggestions, thereby increasing the quality of the manuscript. I recommend the manuscript for punctuation as it stands.